# Adaptive Wireless Image Transmission Transformer Architecture for Image Transmission and Reconstruction

**DOI:** 10.3390/s24216772

**Published:** 2024-10-22

**Authors:** Hongcheng Li, Geming Xia, Chaodong Yu, Yuze Zhang, Hongfeng Li

**Affiliations:** School of Computer Science, National University of Defense Technology, Changsha 410000, China; lihongcheng22@nudt.edu.cn (H.L.); chaodongyu16@nudt.edu.cn (C.Y.); zhangyuze2025@163.com (Y.Z.); lihongfeng22@nudt.edu.cn (H.L.)

**Keywords:** 6G, semantic communication, deep learning, channel adaptation, WITT model

## Abstract

The advancement of 6G (6th Generation Mobile Networks) communication technology has posed challenges for traditional communication network architectures in meeting the demands for communication efficiency and quality. Semantic communication technology, characterized by its “understand before transmit” approach, has emerged as a pivotal technology driving the progress of 6G due to its ability to enhance communication efficiency and quality. The Wireless Image Transmission Transformer (WITT) model, which operates as a semantic communication system leveraging vision transformer technology for the transmission of semantic images, has shown efficacy in transmitting input images through processes of feature extraction and channel adaptation. This study introduces an advanced channel adaptive module that is informed by deep learning methodologies and the adaptive modulation principles of the Variational Information Bottleneck (VIB). This innovation enhances the original WITT model, resulting in the development of the Adaptive Wireless Image Transmission Transformer (ADWITT) architecture. Comprehensive experimental results have unequivocally shown that the transmission performance of the ADWITT architecture substantially surpasses that of the conventional WITT (Wavelet Image Transmission Technique) model, particularly in scenarios characterized by harsh and detrimental channel conditions. These findings underscore the robustness and adaptability of the ADWITT approach, which is poised to improve the field of image transmission by offering superior performance and resilience in environments where traditional methods falter.

## 1. Introduction

The evolution of communication technology has progressed from 1G (1st Generation Mobile Networks) to 5G (5th Generation Mobile Networks), resulting in significant enhancements in communication efficiency by orders of magnitude. System capacity has also been approaching the Shannon limit. However, the emergence of 6G has introduced even higher demands for system capacity and communication efficiency due to the increasing requirements of new applications that necessitate the transmission of larger amounts of data within a constrained bandwidth. Consequently, conventional communication methods are no longer adequate to meet these evolving demands, thereby rekindling interest in semantic communication, which has been previously overlooked. Semantic communication has now become a focal point of current research.

In recent years, the exploration of 6G scenarios has garnered significant attention, particularly in the realm of integrating artificial intelligence (AI) with sensing and communication systems. This trend is comprehensively discussed in the seminal article featured in the *IEEE Communications Magazine*, titled “AI-Enhanced Integrated Sensing and Communications: Advancements, Challenges, and Prospects” [1]. This publication serves as a cornerstone for understanding the profound implications of AI in revolutionizing the functionality and performance of 6G networks.

Drawing inspiration from this foundational work, Chowdary et al. [2] delve deeper into the realm of hybrid radar fusion techniques. Their research highlights the potential of these advanced methodologies to significantly enhance the performance of integrated sensing and communication systems. By leveraging the unique capabilities of hybrid radar fusion, Chowdary and his colleagues demonstrate how these systems can achieve unprecedented levels of accuracy and reliability.

In parallel, Bazzi and Chafii [3] focus on outage-based beamforming designs tailored specifically for dual-functional radar–communication 6G systems. Their work addresses a critical challenge in the field, namely optimizing beamforming strategies to ensure robust performance even in the presence of outages. The authors propose innovative designs that strike a balance between maximizing throughput and minimizing outages, thereby enhancing the overall resilience of 6G networks.

Furthermore, Chafii et al. [4] present a comprehensive overview of twelve scientific challenges facing the development of 6G technology. Their analysis underscores the need for a fundamental reevaluation of traditional communications theory in light of emerging technological trends. The authors advocate for a more holistic approach to addressing these challenges, one that incorporates insights from across various disciplines and leverages cutting-edge advancements in AI, machine learning, and beyond.

Meanwhile, Liu et al. [5] contribute to the ongoing evolution of multiple access techniques by examining the transition from non-orthogonal multiple access (NOMA) to next-generation multiple access (NGMA) for 6G. Their work discusses the potential benefits of NGMA, including increased spectral efficiency, improved user fairness, and reduced latency. Liu and his colleagues propose novel algorithms and architectures that facilitate the seamless integration of NGMA into 6G networks, paving the way for future advancements in this critical area.

Research on semantic communication is still in its nascent stages and is confronted with various challenges, particularly in the realm of encoding and decoding technology. The primary challenges in ongoing research include the integration of semantic communication modules into conventional communication architectures and the development of an adaptive semantic communication framework that enables the system to select suitable encoding and decoding methods based on varying channel conditions.

Collectively, these contributions provide a comprehensive overview of the current state and future directions in 6G scenarios. By exploring innovative approaches such as AI integration, sophisticated beamforming techniques, and advanced multiple access strategies, researchers are laying the groundwork for the development of more efficient, reliable, and resilient 6G networks. As the field continues to evolve, these contributions will undoubtedly serve as valuable resources for guiding future research and development efforts, further enriching the landscape of semantic communication and its integration within the broader framework of 6G technology.

Efforts to integrate semantic communication modules into traditional communication architectures are currently being pursued along three main avenues: incorporating channel semantic encoders and decoders at both ends of the channel, integrating semantic encoders and decoders separately at the source and destination, and deploying semantic encoders and decoders at both the source and destination while also including them at both ends of the channel to achieve a communication effect akin to end-to-end joint training.

Given the extensive research and development in communication systems, it has become increasingly evident that traditional frameworks have reached their theoretical performance limits, particularly when it comes to module separation designs. This realization has sparked renewed interest in joint source–channel coding, which offers significant advantages over traditional approaches. Reference [6] delves into the transition from linear to nonlinear processing in joint source–channel coding, introducing pivotal technologies and two innovative mechanisms: “nonlinear transformation” and “nonlinear coding”. These advancements not only enhance the quality of information exchange in semantic communication systems but also reduce data transmission demands, thereby substantially boosting transmission performance.

Reference [7] further contributes to this field by presenting an efficient joint source–channel coding scheme that leverages nonlinear transformation to closely align with source distribution. Unlike traditional deep joint source–channel coding methods, this innovative scheme learns the latent representation of the source and the entropy model (serving as the prior for the latent representation). As a result, it introduces a novel adaptive rate transmission mechanism and a super-prior-assisted codec refinement strategy, both aimed at improving deep joint source–channel encoding. The overall system is designed to minimize rate–distortion performance using metrics such as PSNR/MS-SSIM and the emerging human perception metric LPIPS, thereby aligning with the broader goals of end-to-end semantic communication. Reference [8] addresses the challenges of module design separation and traditional processing paradigms by proposing a new framework for end-to-end nonlinear transformation encoding in semantic communication. Based on variational theory, this study derives optimization criteria for end-to-end rate–distortion, designs a nonlinear transformation for semantic information extraction, and employs semantic variational entropy modeling for variable-rate joint source–channel coding. Experimental results showcase significant improvements in end-to-end data transmission performance and robustness.

The deep Joint Source–Channel Coding (JSCC) architecture integrates deep learning with semantic communication by introducing a semantic extraction module in the joint encoder, directly mapping input images to channel inputs. Reference [9] evaluates this architecture across AWGN and slow fading Rayleigh channel models, demonstrating superior performance across all average signal-to-noise ratio (SNR) values and outperforming traditional separation-based schemes at any channel bandwidth. Building on this, Reference [10] enhances the deep JSCC architecture with Orthogonal Frequency-Division Multiplexing (OFDM) for wireless image transmission in multipath fading channels, achieving better performance than traditional OFDM-based separate source–channel coding, especially at low SNR and transmission rates.

Furthermore, current research suggests that deep learning can effectively address challenges in the architectural design of semantic communication systems. Reference [11] introduces the Deep Image Semantic Communication model (DeepISC), integrating ViTA and DSN based on vision transformers to tackle imprecision in semantic feature extraction, inaccuracies in image reconstruction, and the lack of relevant evaluation metrics. This model also proposes a comprehensive multi-task image semantic evaluation methodology, tailored to various downstream tasks.

The studies referenced in the literature highlight innovative approaches to semantic communication leveraging deep learning technologies. One study introduces a semantic communication system model specifically designed for intelligent tasks within the Artificial Intelligence Internet of Things (AIoT) context. This model addresses the challenges of traditional centralized cloud computing, such as extensive data transmission, communication delays, and detrimental effects on intelligent task performance. By integrating semantic concepts with feature maps, the proposed architecture facilitates the extraction of semantic relationships and enables feature map compression and transmission, achieving data compression at the semantic level. Experimental results validate the feasibility and superior performance of this model, especially in AIoT environments with a constrained bandwidth and delay conditions [12].

Another study proposes a wireless image transmission system enhanced with multi-level semantic information, utilizing an end-to-end training approach. The encoder of this system consists of a multi-level semantic feature extractor and a joint semantic channel encoder. By incorporating a Channel Attention (CA) module and a pixel up-sampling module, the system learns input features and enhances the dimensionality of feature information. High-level semantic information is then cascaded with low-level detailed supplementary information to extract fused information and reconstruct the target image. The experimental findings indicate significant improvements in the effectiveness and efficiency of this semantic communication system compared to traditional systems, particularly under bandwidth-limited conditions [13].

Furthermore, a novel approach for multi-task image feature extraction and compression using deep learning techniques has been introduced. This approach employs a primary network for extracting high-level textual features, an intermediate network for capturing high-level semantically related image features, and a supplementary network for acquiring low-level pixel detail features. To address the artifact block phenomenon during image reconstruction, a dual-feature module is proposed. The system model is designed to accommodate various downstream tasks by modifying the final layer, and experimental results demonstrate superior Peak Signal-to-Noise Ratio (PSNR) performance in downstream task processing [14].

In parallel with these advancements, researchers are exploring the integration of other machine learning methodologies with semantic communication. One study introduces a new AI-driven communication paradigm that focuses on the exploration, transmission, and recovery of fundamental semantic information to facilitate more efficient communication. A key feature of this paradigm is the propagation of models, and the literature elaborates on the realization of the “intelligent flow” of semantic information through model propagation. The concept of the semantic slice model (SeSM) is introduced to minimize the communication overhead associated with model propagation, enabling flexible model replication tailored to varying requirements concerning model performance, channel conditions, and transmission objectives [15].

Reference [16] introduces a semantic image coding approach for multimedia semantic communication leveraging Generative Adversarial Networks (GANs), aiming to facilitate semantic exchange rather than mere symbol transmission. This work details an image semantic coding model specifically designed for multimedia semantic communication systems, accompanied by an optimization framework focused on balancing pass rate–distortion perception. The model achieves remarkable visual reconstruction and semantic preservation at minimal bit rates, structured into a base layer for complete semantic information retention and an enhancement layer for detailed image restoration. Fine-tuning parameters in both layers achieves a balance between perceptual and distortion performance.

To mitigate training resource consumption in variable channel conditions, current research emphasizes designing a channel adaptive module for deep joint source–channel coding. Reference [11] proposes an adaptive semantic coding method using reinforcement learning for bandwidth-sensitive, semantic-rich image data communication systems. This method transcends pixel-level encoding and introduces a GAN-based semantic decoder integrating local and global features through an attention module, adapting the semantic communication model to channel conditions. Experimental results showcase robustness to noise and high-quality image reconstruction at low bit rates.

Building on this, study [17] investigates a channel adaptive design for deep joint source–channel coding to optimize performance under varying channel conditions while minimizing resources. A novel lightweight attention-based adaptive feature (AF) module is introduced, generating scaling sequences for feature sets across different signal-to-noise ratios (SNRs), enabling channel-level feature recalibration. The architecture alternates feature learning (FL) modules with attentional feature (AF) modules, allowing dynamic feature scaling according to channel state for self-adaptation.

Furthermore, to enhance image transmission quality in low-SNR channels, the work presented in [18] introduces a mechanism for channel representation learning and adaptive modulation tailored for semantic communication. Using semantic segmentation, it encodes objects and backgrounds separately, prioritizing transmission of classification-critical object components under low-SNR conditions while safeguarding background information for higher SNRs. Empirical evaluations demonstrate improved classification accuracy and mean square error performance across various SNR levels.

Expanding on prior research, study [19] introduces the Wireless Image Transmission model for semantic communication, known as the WITT (Wireless Image Transmission Transformer) model. This model is specifically developed for transmitting high-resolution images by leveraging convolutional neural networks (CNNs). To overcome challenges related to capturing global features efficiently and thereby enhancing end-to-end transmission performance, the model integrates Swin Transformer modules for superior feature extraction quality. Additionally, a channel adaptive module is proposed to adjust extracted semantic features based on channel state information, thereby improving the model’s performance under varying channel conditions. Empirical evaluations demonstrate the effectiveness of the WITT model in maintaining high performance levels across different image resolutions, distortion indices, and channels, surpassing conventional architectures significantly.

This study extends the existing Wireless Image Transmission and Reconstruction Technique (WITT) model by significantly enhancing its channel adaptive module through the integration of pertinent principles from deep learning. This integration has led to the development of an advanced architecture, termed ADWITT (Adaptive Deep WITT), specifically tailored for image transmission and reconstruction. The ADWITT architecture leverages the high-performing Swin Transformer module from the original WITT model, harnessing its capabilities for superior feature extraction from input images.

To address the limitations of the original WITT model in complex channel environments, this paper presents a redesign of the channel adaptive module. This newly designed module incorporates self-attention mechanisms, enabling sophisticated modulation of input features. This innovation ensures high-quality image reconstruction at the receiving end, even under complex and challenging channel conditions, without the necessity for retraining. Consequently, the final reconstructed images demonstrate performance levels that are comparable to the original images across a wide range of complex channel scenarios.

This study showcases the substantial performance benefits of the ADWITT architecture, as outlined through rigorous research and comparative experiments. The results indicate notable enhancements in performance metrics, including the Peak Signal-to-Noise Ratio (PSNR) and Multi-Scale Structural Similarity Index (MS-SSIM) [20], particularly in challenging channel environments. These findings demonstrate a substantial improvement in performance compared to the conventional WITT model. As channel conditions worsen, the advantages of the ADWITT architecture become even more pronounced, highlighting its robustness and adaptability in diverse and complex transmission environments.

## 2. The Structure of the Model Architecture

### 2.1. Overall Structure

The proposed ADWITT model architecture, as illustrated in Figure 1, comprises a joint source–channel encoder that integrates an efficient feature extraction module utilizing the Swin Transformer module and a channel adaptation module capable of adjusting to varying channel conditions.

To elaborate, the initial image X undergoes processing by the feature extraction module to generate image feature X’, which is subsequently transmitted through the channel adaptive module and wireless channel. The feature extraction module is composed of multiple stages, each consisting of a down-sampling patch merging layer and several Swin Transformer blocks. The configuration of stages and the depth of Swin Transformer blocks within each stage are tailored based on the input image resolution; typically, higher image resolutions necessitate more stages and deeper Swin Transformer blocks within each stage. Previous studies [19] have demonstrated the efficacy of this feature extraction module in efficiently extracting features from input images, particularly for high-resolution images. Post-feature extraction, this module excels in reconstructing intricate image details at the receiving end, thereby significantly enhancing the overall model architecture’s performance.

Subsequently, the extracted image features X’ are fed into the channel adaptive module for adaptive modulation, and the processed image features are transmitted through the wireless channel after traversing the power normalization layer. The primary role of the power normalization layer is to ensure that the image features X* post-encoder processing adhere to average power constraint conditions. This research examines two prevalent fading channel models: the Additive White Gaussian Noise (AWGN) channel and the Rayleigh fast fading channel. The transfer function of these channels is denoted as
(1)Y*=N(X*,K)=K⨀X*+B
where ⨀ signifies the product of elements, K represents the channel state information (CSI) vector, and each element of the noise vector B conforms to a Gaussian distribution
(2)$$\left(B\sim N(0,\sigma_{n}^{2}\cdot I_{k})\right)$$
with the average noise power denoted as $\sigma_{n}^{2}$.

The decoder component of the model’s source–channel joint decoder exhibits a structure akin to that of the encoder. It undertakes the task of reconstructing image Y from a noisy information-laden image feature Y* using multiple modules. The Mean Squared Error (MSE) is computed by comparing the original image X with the reconstructed image Y. Peak Signal-to-Noise Ratio (PSNR) and Multi-Scale Structural Similarity (MS-SSIM) metrics are employed for fine-tuning the overall model architecture. The model is trained end-to-end, enabling the adjustment of parameters within each module to optimize the model’s performance. This approach significantly bolsters the model’s adaptability and resilience.

### 2.2. Channel Adaptive Module

The channel adaptive module introduced in this study is designed to dynamically adjust modulation modes based on real-time channel conditions. Utilizing a variational approximation method inspired by the Information Bottleneck (IB) principle, as previously detailed in [21], and incorporating both multiplicative and additive modulation techniques outlined in [22], we have enhanced the Channel Modnet module within the original WITT model. These enhancements allow the module to select optimal modulation strategies in response to varying channel conditions, significantly improving the adaptability of the semantic communication system model. The architecture of this module is illustrated in Figure 2.

We introduce a variational approximation methodology rooted in the Information Bottleneck (IB) principle, designed to adaptively employ diverse modulation strategies contingent upon channel conditions. In scenarios characterized by favorable channel conditions, specifically a high signal-to-noise ratio (SNR), we draw inspiration from direct modulation techniques to optimize transmission efficiency while safeguarding high-fidelity image transfer. Conversely, under adverse channel conditions marked by a low SNR, we harness more intricate modulation paradigms to conserve additional image details, thereby augmenting image transmission quality.

To further hone this approach, we refrain from directly implementing specific modulation techniques such as Amplitude Modulation (AM) in analog modulation, Frequency Modulation (FM), and Phase Modulation (PM) in digital modulation. Instead, we deeply integrate the fundamental principles of these modulation techniques. At the multiplicative modulation level, inspired by AM, we derive a modulation factor contingent upon channel conditions and multiply it by image features. This process transcends a mere replication of AM, embodying an intelligent modulation of image features at the feature level, thereby indirectly adjusting the amplitude characteristics of the feature map to ensure image information aligns with the instantaneous channel state for transmission.

Furthermore, in the context of complex channel conditions, we draw inspiration from FM and PM techniques to innovatively generate auxiliary modulation factors aimed at amplifying and enriching the expressive capability of image details. The deployment of these auxiliary modulation factors does not mirror the modulation processes of FM or PM directly but subtly adjusts the frequency and phase characteristics of the feature map through manipulation of image features at these respective levels. This methodology is particularly pivotal in complex channel environments, as it maintains core image information while effectively preserving and enhancing image details.

The channel adaptive module (CAM) undertakes a sophisticated modulation process as outlined below: Upon receiving image features extracted by an advanced feature extractor, the CAM begins by assessing the current channel conditions in real-time. Based on these assessments, the bm-list module primarily applies Amplitude Modulation (AM) principles to derive a multiplicative modulation factor (bm). This bm factor is dynamically computed and then multiplied with the modulated image features generated by the preceding sm-list module, resulting in the creation of refined hidden features termed temp1. As these hidden features traverse subsequent sm-list and bm-list modules within the CAM, they undergo further enhancement and modulation.

Each sm-list module consists of a linear layer followed by a Rectified Linear Unit (ReLU) activation layer. This architectural design facilitates enhanced processing and amplification of image features, thereby contributing to the robustness and quality of the transmitted signal. Conversely, the bm-list module’s primary responsibility is to generate the bm factor, which acts as a multiplicative scaling factor for the modulated image features. This scaling process is pivotal in maintaining signal quality and stability during transmission, ensuring that the modulation strategy aligns optimally with the prevalent channel environment.

The bm-list module’s AM structure is designed for maximum adaptability. It comprises three linear layers interspersed with two ReLU activation layers, enabling the module to learn and apply intricate transformations to the image features. The input image features undergo a series of linear transformations, each followed by a ReLU activation function. The ReLU function introduces nonlinearity, enabling the model to capture and learn more sophisticated patterns within the data. Upon traversing all three linear layers and two ReLU activation layers, the bm-list module outputs the bm factor, which is then utilized to scale the modulated image features and produce the refined hidden features (temp1).

In challenging channel conditions, the bm-list module transitions to a more advanced modulation structure known as Hybrid Amplitude Modulation (HAM). HAM combines the outputs of two AM structures, each with distinct linear layers and ReLU activation layers, to generate a more resilient bm factor. This enhanced bm factor offers superior adaptation to noisy and interference-prone channel conditions, further bolstering the quality and stability of the transmitted signal.

In summary, the bm-list module within the CAM dynamically selects the appropriate modulation structure (AM or HAM) based on real-time channel condition assessments. This adaptive modulation strategy ensures that image features are effectively scaled and transmitted with minimal distortion, even in complex and dynamic communication environments. By preserving intricate image details, the CAM elevates the quality of reconstructed images at the receiving end, thereby fulfilling the demands of downstream tasks such as image recognition, analysis, and processing.

After passing through all sm-list and bm-list modules, the resultant hidden feature (temp) is fed into a Sigmoid activation layer. This layer generates a modulation factor (mod-val) within the range of 0 and 1, which is then multiplied with the original image features to produce the output image features. Notably, in situations where channel quality significantly diminishes, the cm-list module within the CAM generates an additive modulation factor (cm) based on current channel conditions. This factor is added to the previously modulated image features to yield the final output image features.

By dynamically selecting appropriate modulation modes—whether multiplicative (utilizing AM and HAM principles) or additive (cm)—based on varying channel conditions, the CAM effectively mitigates noise interference in image features during wireless transmission. Consequently, it enhances the quality of reconstructed images at the receiving end and addresses the issue of excessive image distortion prevalent in traditional communication systems under complex channel conditions. Furthermore, the CAM preserves intricate image details, ensuring that the quality of reconstructed images meets the stringent requirements of downstream tasks.

## 3. Experimental Results and Analysis

### 3.1. Preparation of Experimental Conditions

Datasets: This study utilized various datasets of different resolutions to train and assess the ADWITT architecture introduced in this research. CIFAR10 datasets [23] were chosen for training and testing with low-resolution images, while DIV2K datasets [24] were used for training and Kodak datasets [25] for testing with high-resolution images.

Contrast experiment: In the comparative experiment, the ADWITT architecture was evaluated against the WITT model [19], as documented in the existing literature, alongside the conventional deep Joint Source and Channel Coding (JSCC) architecture [9] and traditional methods of separate source and channel coding. For the purpose of this study, BPG coding [26] was employed as the source coding technique, while 5G Low-Density Parity-Check (LDPC) coding [27] was utilized for channel coding. The research seeks to assess the performance of these architectures across various resolution datasets and channel conditions, with the objective of elucidating differences in transmission performance under diverse scenarios.

Channel model selection: Two channel models, namely the AWGN channel and RAYLEIGH fast fading channel, were selected to assess the performance of the three architectures, testing the model’s adaptability.

Evaluation index: Given that the ADWITT model presented in this study is specifically tailored for image reconstruction tasks, it is imperative that the performance evaluation metric aligns closely with the requirements of such tasks. Consequently, this paper selects Mean Squared Error (MSE) as the training metric, while evaluation of test outcomes is conducted using Peak Signal-to-Noise Ratio (PSNR) and Multi-Scale Structural Similarity (MS-SSIM) metrics [28]. The PSNR is determined by the formula
(3)$$PSNR=10*\log_{10}\left({MAX}^{2}/MSE\right)$$
where MAX denotes the maximum pixel value of the image. A higher PSNR value signifies better image quality post-processing. On the other hand, MS-SSIM is calculated as
(4)$$MS-SSIM(X,Y)={\left[L_{M}(X,Y)\right]}^{\alpha M}\cdot \prod_{j=1}^{M}{\left[C_{j}(X,Y)\right]}^{\beta j}{\left[S_{j}(X,Y)\right]}^{\gamma j}$$
with X and Y representing the images under comparison. $L_{M}(X,Y)$ reflects the brightness contrast on the largest scale, while $C_{j}(X,Y)$ and $S_{j}(X,Y)$ denote the contrast and structural comparisons on the jth layer scale. Parameters α,β,γ are employed to adjust the weightage of different components. The MS-SSIM value ranges between 0 and 1, indicating the structural similarity between the original and reconstructed images. A value closer to 1 signifies a higher structural resemblance. However, as the MS-SSIM values in this study are close to 1, they are converted to decibel (dB) units for easier comparison.

Training details: During training, SNR values ranging from 0 dB to 20 dB were randomly generated based on the WITT model. The channel adaptive module selection was contingent on the SNR value, with the AM architecture chosen as the bm-list module when the SNR exceeded 10. For SNR values below 10, the HAM architecture was preferred, and for SNR values below 5, the cm-list module was utilized for additive modulation. An end-to-end training approach was adopted to synchronize each module within the training architecture. The specific parameter settings are shown in Table 1.

### 3.2. Analysis of Experimental Results

Figure 3 illustrates the correlation between Peak Signal-to-Noise Ratio (PSNR) values achieved through various coding modes and signal-to-noise ratios (SNRs) across different channel conditions within datasets characterized by low resolution. The findings indicate that the ADWITT architecture proposed in this study outperforms the conventional WITT model, deep Joint Source–Channel Coding (JSCC) architecture, and discrete source–channel coding in AWGN channels, exhibiting higher PSNR values across all SNR levels. Furthermore, in the context of RAYLEIGH fast fading channels, the ADWITT structure demonstrates considerable performance advantages over the aforementioned three methods, particularly under low SNR conditions. At elevated SNR levels, the ADWITT architecture exhibits performance that is comparable to traditional split source–channel coding, while significantly surpassing the performance of the other two coding techniques. It is important to note that the performance of all architectures is diminished in RAYLEIGH fast fading channels when compared to AWGN channels. This degradation is attributed to the severe distortion and errors in the received signal caused by rapid fading and multipath effects inherent in the channel. Conversely, in AWGN channels, the distortion and errors are relatively minimal, which contributes to the overall performance reduction of each model in RAYLEIGH fast fading channels. These results underscore the capability of the ADWITT architecture within datasets characterized by low resolution to sustain high image quality across varying SNR conditions and to preserve the similarity between the transmitted image and the original image, regardless of whether the channels are AWGN or RAYLEIGH fast fading.

Figure 4 depicts the relationship between the Peak Signal-to-Noise Ratio (PSNR) values achieved through various encoding modes in high-resolution image scenarios and the signal-to-noise ratio (SNR) across different channel conditions. The findings indicate that the performance of the ADWITT structure introduced in this study surpasses that of the conventional WITT model, deep Joint Source–Channel Coding (JSCC) framework, and separated source–channel coding, particularly under the condition of AWGN channels with low-SNR conditions. In high-SNR scenarios, the ADWITT architecture consistently maintains the performance of the optimal model and demonstrates a significant advantage over the deep JSCC architecture. Furthermore, in the RAYLEIGH fast fading channel environment, the ADWITT structure consistently upholds the performance of separated source–channel coding under low-SNR conditions, outperforming both the traditional WITT model and the deep JSCC framework. In high-SNR conditions, the ADWITT structure consistently matches the performance of the WITT model and separated source–channel coding, while also exhibiting a marked improvement over the deep JSCC architecture. The experimental results affirm that the ADWITT architecture proposed in this study is capable of achieving high-quality image reconstruction across all SNR conditions when processing high-resolution images, thereby rendering the ADWITT model more adaptable and robust in accommodating various channel conditions.

Figure 5 illustrates the correlation between the MS-SSIM values derived from various encoding modes for low-resolution images and the signal-to-noise ratio (SNR) across different channel conditions. The findings indicate that the performance of the ADWITT architecture presented in this study significantly surpasses that of the other three codec schemes across all SNR ranges, regardless of whether in AWGN channels or Rayleigh fast fading channels. This suggests that the ADWITT architecture is capable of achieving high-quality image reconstruction under all SNR and channel conditions when processing low-resolution images. Furthermore, it demonstrates that the images reconstructed by the ADWITT architecture exhibit greater alignment with the perceptual characteristics of the human visual system, particularly in terms of brightness, contrast, and structural fidelity.

Figure 6 illustrates the correlation between the MS-SSIM values derived from the four codec methods for high-resolution images and the signal-to-noise ratio (SNR) across various channel conditions. The findings indicate that, in the context of AWGN channels with a low SNR, the ADWITT architecture proposed in this study exhibits performance comparable to that of the deep JSCC architecture and the traditional separated source–channel codec approach, while significantly outperforming the conventional WITT model. In scenarios characterized by a high SNR, the ADWITT model maintains performance levels akin to those of the traditional WITT model and demonstrates superior efficacy compared to the other two coding methods. Furthermore, under conditions of RAYLEIGH fast fading channels, the ADWITT architecture consistently exhibits performance similar to that of the traditional WITT model across all SNR ranges, while also outperforming the other two codec schemes. The experimental results substantiate that the ADWITT architecture can reconstruct images of higher quality across diverse channel and SNR conditions, thereby affirming the strong applicability and robustness of the proposed ADWITT model.

Table 2 presents a comprehensive evaluation of the inference time, computational complexity, and parameter count associated with the ADWITT, WITT, and ADJSCC [17] methods on the CIFAR-10 dataset, with a batch size of 128 and various signal-to-noise ratio (SNR) values. All experimental assessments were conducted utilizing an NVIDIA RTX 3080 GPU. The table reveals that our proposed ADWITT model exhibits a reduced computational complexity compared to the conventional WITT model. However, this reduction is accompanied by a slight increase in inference time and a corresponding augmentation in the number of parameters, albeit within tolerable limits. Furthermore, when benchmarked against the traditional ADJSCC model, our ADWITT model demonstrates a faster inference time and lower computational complexity, despite the increase in parameter count. These findings indicate that our ADWITT model has achieved a noticeable enhancement in performance.

## 4. Conclusions

To enhance the adaptability of the traditional WITT model across various channel conditions, the channel adaptive module within the model architecture has been enhanced, leading to the development of the novel ADWITT architecture for image transmission and reconstruction. By adapting modulation modes based on the channel conditions, the ADWITT architecture can consistently ensure high-quality transmission and reconstruction of image data in complex channel environments. Experimental results confirm the robust performance of the ADWITT architecture across multiple channel models and diverse SNR conditions, thereby enhancing its applicability and resilience. This architecture can provide new ideas and methods for future research on high-speed and high-precision semantic communication.

## Figures and Tables

**Figure 1 sensors-24-06772-f001:**
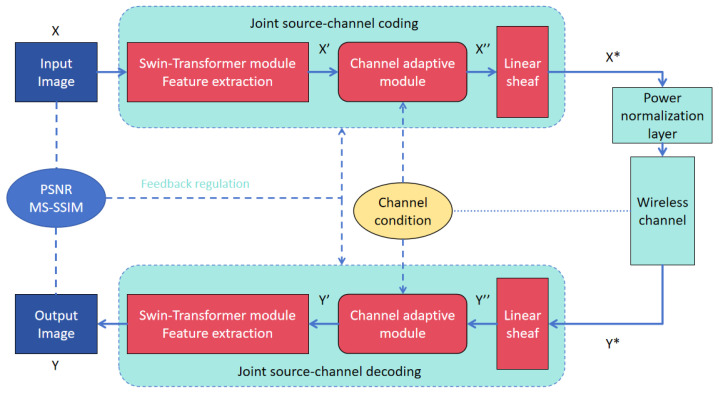
The comprehensive structure of ADWITT.

**Figure 2 sensors-24-06772-f002:**
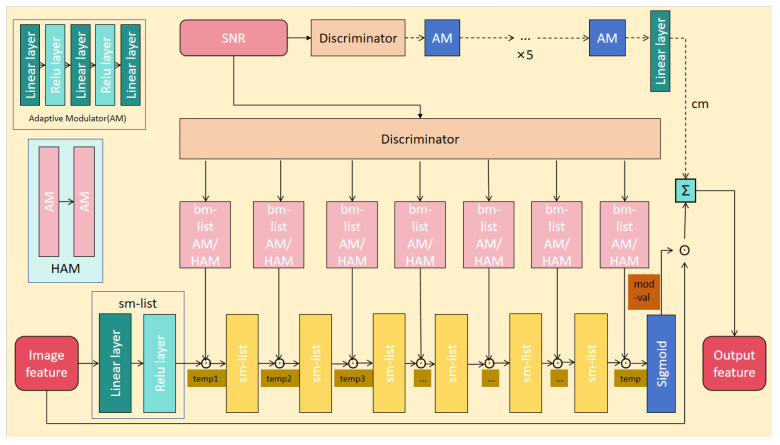
The structure of the channel adaptive module.

**Figure 3 sensors-24-06772-f003:**
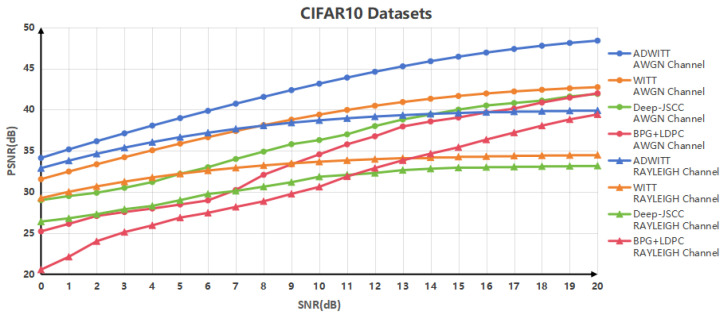
PSNR performance of CIFAR10 datasets in AWGN channel and RAYLEIGH fast fading channel.

**Figure 4 sensors-24-06772-f004:**
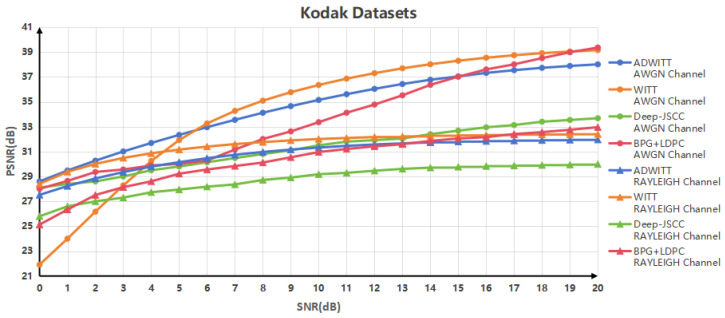
PSNR performance of Kodak datasets in AWGN channel and RAYLEIGH fast fading channel.

**Figure 5 sensors-24-06772-f005:**
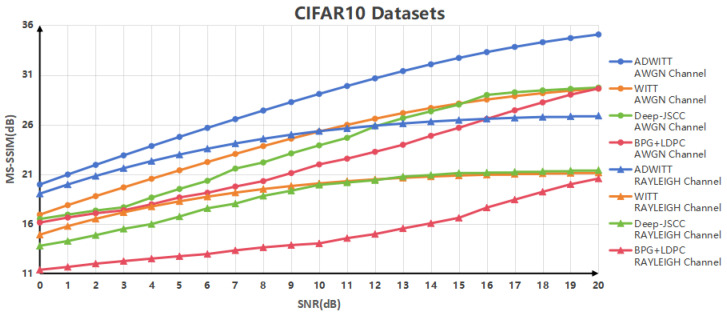
MS-SSIM performance of CIFAR10 datasets in AWGN channel and RAYLEIGH fast fading channel.

**Figure 6 sensors-24-06772-f006:**
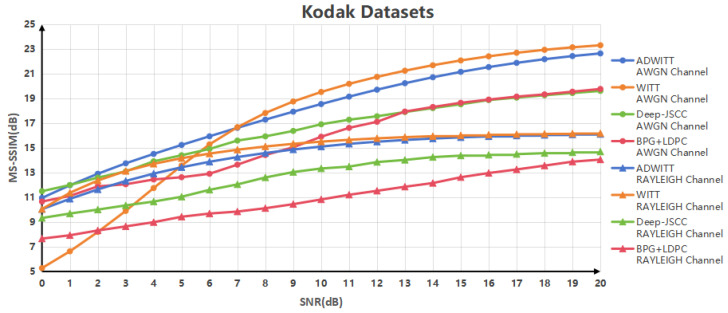
MS-SSIM performance of Kodak datasets in AWGN channel and RAYLEIGH fast fading channel.

**Table 1 sensors-24-06772-t001:** Parameter settings.

Image Resolution and Datasets	The Configuration of Stages	The Depth of Swin Transformer Blocks	Window Size	Batch Size	Learning Rate
Low image resolution (CIFAR 10 datasets)	2	[2,4]	2	128	$10^{-4}$
High image resolution (DIV2K datasets and Kodak datasets)	4	[1,1,2,6]	8	16	$10^{-4}$

**Table 2 sensors-24-06772-t002:** Resource loss parameter.

Method	SNR	Inference Time	Computational Complexity	Number of Parameters
ADWITT	3 dB	0.1250 ms	727.17 MMac	21.22 M
	8 dB	0.1215 ms	725.79 MMac	21.22 M
	15 dB	0.1197 ms	719.56 MMac	21.22 M
WITT	3 dB	0.1143 ms	775.97 MMac	13.8 M
	8 dB	0.1141 ms	775.97 MMac	13.8 M
	15 dB	0.1134 ms	775.97 MMac	13.8 M
ADJSCC	3 dB	0.1532 ms	2065.21 MMac	7.93 M
	8 dB	0.1518 ms	2065.21 MMac	7.93 M
	15 dB	0.1514 ms	2065.21 MMac	7.93 M

## Data Availability

The data presented in this study are available on request from the corresponding author.

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
