# Peer review of "Adaptive Wireless Image Transmission Transformer Architecture for Image Transmission and Reconstruction"

_sensors, 2024, doi:10.3390/s24216772_

Round 1
Reviewer 1 Report (New Reviewer)
Comments and Suggestions for Authors
The article developed a novel ADWITT architecture with the channel adaptive module to enhance the adaptability of the traditional WITT model across various channel conditions for image transmission and reconstruction. It could be accepted after the revisions listed below are fulfilled.
1.Please ensure all the abbreviations are well-defined before using them, such as 1G, 5G, 6G.
2.It is strongly recommended to list the contributions of the paper, since the description of the current version is not particularly clear.
3.The number of references is too small, resulting in insufficient literature research, although the length of the introduction is quite long. It is suggested to refine the introduction related to this topic. It would be better if the authors could add some discussions on how extend this work on recent 6G scenarios such as "AI-Enhanced Integrated Sensing and Communications: Advancements, Challenges, and Prospects", IEEE Communications Magazine, 2024.
4.Please clarify the difference between the proposed design and the traditional WITT model.
5.If possible, please add more necessary theoretical analysis in the paper.
6.Table 1 should be re edited.
Comments on the Quality of English LanguageMinor editing of English language required.
Author Response
Dear Reviewer,
I would like to extend my heartfelt thanks for taking the time to review our manuscript and for providing valuable feedback that has greatly contributed to its improvement. I am pleased to inform you that I have carefully addressed all of your comments and questions, responding to each point in detail in the attached document. Please find it enclosed and review it at your convenience. Thank you once again for your insightful input.
Sincerely,
Hongcheng Li

Reviewer 2 Report (New Reviewer)
Comments and Suggestions for Authors
The paper introduces the Adaptive Wireless Image Transmission Transformer (ADWITT) architecture, an enhancement of the existing Wireless Image Transmission Transformer (WITT) model, where objective is to improve the efficiency and quality of image transmission in 6G communication systems by incorporating advanced deep learning and adaptive modulation strategies - paper also explores semantic communication concepts and proposes an adaptive channel model to achieve high-quality image reconstruction.
My comments follow:
1. Many missing 6G works in the introduction should be included like [R1,R2,R3,R4].
2. Provide more discussion on the computational complexity introduced by the adaptive modulation techniques, especially in real-time implementations especially when you compare with classical wireless image transmission techniques.
3. Explain the trade-offs between performance improvement and computational resource utilization.
4. The modulation factors (e.g., amplitude modulation and frequency modulation) are selected and applied is somewhat dense and could benefit from a more accessible explanation for readers less familiar with advanced modulation techniques.
5. The paper does not provide a comprehensive comparison of ADWITT’s performance in terms of computational overhead compared to other models. Including these details would provide readers with a clearer picture of the trade-offs involved.
6. The paper focuses predominantly on image quality metrics. It would be beneficial to include a broader set of performance indicators, such as latency or throughput, to further emphasize the practicality of ADWITT in real-world 6G systems.
7. The ADWITT model shows results under low SNR conditions, but how good is the system when applied to real-world environments with larger image resolutions or higher densities of communication users?
8. Can this model be used for imaging in integrated sensing and communication systems like in [R2] through wireless signals ?
9. Why were these models not included in the comparative analysis?
10. Some typos appearing need to be corrected like "structured with" should be "composed of”. Please redraft the paper.
Reference
[R1] A. Chowdary, et al. “On Hybrid Radar Fusion for Integrated Sensing and Communication,” in IEEE Transactions on Wireless Communications, doi: 10.1109/TWC.2024.3357573
[R2] A. Bazzi and M. Chafii, “On Outage-Based Beamforming Design for Dual-Functional Radar-Communication 6G Systems,” in IEEE Transactions on Wireless Communications, vol. 22, no. 8, pp. 5598-5612, Aug. 2023, doi: 10.1109/TWC.2023.3235617
[R3] M. Chafii, et al. “Twelve Scientific Challenges for 6G: Rethinking the Foundations of Communications Theory," in IEEE Communications Surveys & Tutorials, vol. 25, no. 2, pp. 868-904, Secondquarter 2023, doi: 10.1109/COMST.2023.3243918.
[R4] Y. Liu et al., "Evolution of NOMA Toward Next Generation Multiple Access (NGMA) for 6G," in IEEE Journal on Selected Areas in Communications, vol. 40, no. 4, pp. 1037-1071, April 2022, doi: 10.1109/JSAC.2022.3145234.
Comments on the Quality of English Language
The English quality is average but can be improved. Please revise and redraft the paper.
Author Response
Dear Reviewer,
I would like to extend my heartfelt thanks for taking the time to review our manuscript and for providing valuable feedback that has greatly contributed to its improvement. I am pleased to inform you that I have carefully addressed all of your comments and questions, responding to each point in detail in the attached document. Please find it enclosed and review it at your convenience. Thank you once again for your insightful input.
Sincerely,
Hongcheng Li

Reviewer 3 Report (New Reviewer)
Comments and Suggestions for Authors
I find section 2 (the description of the system) quite obscure and dificult to follow and understand. Particularly, I was completely puzzled by section 2.2. Maybe it is because I am not an expert in this kind of systems, but I have to say that I think because it is not clearly explained. Most of the submodules are not properly described (neither their functionality nor how exactly are they implemented).
For example, assertions such as "In particular, we draw upon the amplitude modulation (AM) technique from analog modulation to derive a modulation factor that is responsive to channel conditions, which is then multiplied with the transmitted image." or "Additionally, if required, we incorporate frequency modulation (FM) and phase modulation (PM) from digital modulation to generate an auxiliary modulation factor aimed at enriching the image details" are quite surprising. I have had a look at [18] which is cited as the main reference source for this adaptative modulator, and the architecture described there is much more clear and the modulation is a digital 64-QAM modulation.
Another puzzling sentence "Notably, when the channel quality is notably degraded, the cm-list module within the channel adaptive module generates an additive modulation factor based on the channel conditions. This factor is added to the previous image features to produce the ultimate output image features."
Neither details are provided about the simulator/test system implementation nor any code is released publicly so the experiments can be reproduced and the results validated.
Table 1 is out of the limit of the margins and can not be completely read.
Comments on the Quality of English Language
English language is mostly fine.
Author Response
Dear Reviewer,
I would like to extend my heartfelt thanks for taking the time to review our manuscript and for providing valuable feedback that has greatly contributed to its improvement. I am pleased to inform you that I have carefully addressed all of your comments and questions, responding to each point in detail in the attached document. Please find it enclosed and review it at your convenience. Thank you once again for your insightful input.
Sincerely,
Hongcheng Li

Round 2
Reviewer 2 Report (New Reviewer)
Comments and Suggestions for Authors
The authors have addressed my comments.
Reviewer 3 Report (New Reviewer)
Comments and Suggestions for Authors
The authors have made some minor changes to the paper, but they still have the same issues I pointed out before. I am sorry but still I can't make anything out of what is stated in section 2.2. Whereas before, the authors claimed they used an AM analog modulation (without providing further information), now they state that they refrain from implementing an specific modulation technique but instead they "deeply integrate the fundamental principles of these modulation techniques". They have expanded and rephrased this section but still, nothing explained there have any sense for me.
As I mentioned in my previous revision, I recommend the paper to be reviewed by someone else. I don't know if there are more reviewers assigned. If they are OK with publishing the paper, I am not against their opinion. But in case there are not other reviewers with favorable reviews, I wouldn't publish the paper as it is right now.
This manuscript is a resubmission of an earlier submission. The following is a list of the peer review reports and author responses from that submission.
Round 1
Reviewer 1 Report
Comments and Suggestions for Authors
In this paper, the authors propose a ADWITT architecture to enable high-quality image reconstruction in complex channel conditions. Simulation results are given.
1. What’s the full name of WITT and ADWITT, there is no presentation in the Abstract and Title.
2. The main contribution should be summarized in the Abstract instead of the background introduction.
3. In the related works, the authors should give more attention to similar works solving the same problem. As far as I know, there are a lot of studies.
4. In the channel adaptive model, how to distinguish the channel conditions? To be honest, the reviewer cannot understand the process and principle of channel adaptive module. The author should explain it more clearly.
5. In the experimental results part, the authors should add a table to summarize the setting parameters. More related schemes should be added as the comparison not only WITT, Deep-JSCC.
6. Although there are many results, but it hard to compare them. Same measurements in Figures can be put in one figure to easily the different performance in different conditions.
Comments on the Quality of English LanguageThere is still space for improvement
Author Response
Thank you very much for taking time out of your busy schedule to review the original work and put forward valuable revision suggestions. I have revised the original draft according to your suggestions. Please see the attachment for the specific modification plan.

Reviewer 2 Report
Comments and Suggestions for Authors
In this paper, the authors proposed a ADWITT architecture, where the adaptive modulation from VIB is combined with the WITT model. Here are some comments:
1. Some better vision models have emerged in recent years, such as ConvNeXt, SwinV2, BEiT, SAM, and CoAtNet. It is suggested to include these model as benchmarks.
2. The channel adaptive module relies on things like channel state information (CSI) and SNR to select the modulation method, but this information may not be completely accurate or in real time, which can affect system decisions.
3. Whether the scalability of the module will be limited under more complex or variable channel conditions, especially under different application scenarios or higher communication frequencies, and whether the module will need to be redesigned or fine-tuned.
4. Why do you choose PSNR and MS-SSIM for fine-tuning the model architecture? Why not the SINR or achievable rate?
5. At high SNR region, the proposed ADWITT model slightly inferior to WITT, not 'align with' as stated in Fig. 5 and Fig. 6.
6. In the abstract part, explanations of some abbreviations are missing, such as WITT and ADWITT. There is a formatting problem with the captions to Figure 1 and 2.
Comments on the Quality of English LanguageThe English expression needs moderately improved.
Author Response

(The authors gave the same response as above.)

Reviewer 3 Report
Comments and Suggestions for Authors
To enhance the adaptability of the traditional WITT model across various channel conditions, the channel adaptive module within the model architecture has been enhanced, leading to the development of the novel ADWITT architecture for image transmission and reconstruction. By adapting modulation modes based on the channel conditions, the ADWITT architecture can consistently ensure high-quality transmission and reconstruction of image data in complex channel environments. Experimental results confirm the robust performance of the ADWITT architecture across multiple channel models and diverse SNR conditions, thereby enhancing its applicability and resilience. This architecture can provide new ideas and methods for future research on highspeed and high-precision semantic communication. The reviewer recommends accepting this manuscript.
Author Response
Dear Reviewer:
First of all, please allow me to take this opportunity to express my deepest gratitude to you. I am deeply honored and encouraged by your positive comments on my submission and your suggested reception. Your affirmation is not only a great recognition of my research work, but also an important incentive for me to continue to explore this field in depth.
At the same time, I would like to thank your journal and editorial team for giving me the opportunity to submit this paper, as well as the professionalism and efficiency shown in the whole review process. I know that the publication of the article is inseparable from the hard work and efforts of every participant, and this honor belongs to all of us.
Thank you again for your support and encouragement! I look forward to receiving your guidance and help in my future academic career.
Yours,
Hongcheng Li
Round 2
Reviewer 2 Report
Comments and Suggestions for Authors
The authors have fully considered my former suggestions. In this case, I'd like to accept it for publication in its current form.